# Inhibition of MELK Protooncogene as an Innovative Treatment for Intrahepatic Cholangiocarcinoma

**DOI:** 10.3390/medicina56010001

**Published:** 2019-12-18

**Authors:** Antonio Cigliano, Maria Giulia Pilo, Marta Mela, Silvia Ribback, Frank Dombrowski, Giovanni Mario Pes, Antonio Cossu, Matthias Evert, Diego Francesco Calvisi, Kirsten Utpatel

**Affiliations:** 1Institut für Pathologie, Universitätsklinikum Regensburg, 93053 Regensburg, Germany; Matthias.Evert@ukr.de (M.E.); Diego-Francesco.Calvisi@klinik.uni-regensburg.de (D.F.C.); kirsten.utpatel@ukr.de (K.U.); 2Institut für Pathologie, Universitätsmedizin Greifswald, 17475 Greifswald, Germany; giuliapilo1983@gmail.com (M.G.P.); melarta@tiscali.it (M.M.); silvia.ribback@uni-greifswald.de (S.R.); frank.dombrowski@uni-greifswald.de (F.D.); 3Department of Clinical, Surgical, Experimental Sciences, University of Sassari, 07100 Sassari, Italy; gmpes@uniss.it (G.M.P.); cossu@uniss.it (A.C.)

**Keywords:** intrahepatic cholangiocarcinoma, MELK, FOXM1, EZH2, targeted therapies

## Abstract

*Background and Objectives*: Intrahepatic cholangiocarcinoma (iCCA) is a pernicious tumor characterized by a dismal outcome and scarce therapeutic options. To substantially improve the prognosis of iCCA patients, a better understanding of the molecular mechanisms responsible for development and progression of this disease is imperative. In the present study, we aimed at elucidating the role of the maternal embryonic leucine zipper kinase (MELK) protooncogene in iCCA. *Materials and Methods*: We analyzed the expression of MELK and two putative targets, Forkhead Box M1 (FOXM1) and Enhancer of Zeste Homolog 2 (EZH2), in a collection of human iCCA by real-time RT-PCR and immunohistochemistry (IHC). The effects on iCCA growth of both the multi-kinase inhibitor OTSSP167 and specific small-interfering RNA (siRNA) against *MELK* were investigated in iCCA cell lines. *Results*: Expression of MELK was significantly higher in tumors than in corresponding non-neoplastic liver counterparts, with highest levels of MELK being associated with patients’ shorter survival length. In vitro, OTSSP167 suppressed the growth of iCCA cell lines in a dose-dependent manner by reducing proliferation and inducing apoptosis. These effects were amplified when OTSSP167 administration was coupled to the DNA-damaging agent doxorubicin. Similar results, but less remarkable, were obtained when *MELK* was silenced by specific siRNA in the same cells. At the molecular level, siRNA against *MELK* triggered downregulation of *MELK* and its targets. Finally, we found that MELK is a downstream target of the E2F1 transcription factor. *Conclusion*: Our results indicate that MELK is ubiquitously overexpressed in iCCA, where it may represent a prognostic indicator and a therapeutic target. In particular, the combination of OTSSP167 (or other, more specific MELK inhibitors) with DNA-damaging agents might be a potentially effective therapy for human iCCA.

## 1. Introduction

Cholangiocarcinoma (CCA) is a heterogeneous group of highly aggressive epithelial malignancies originating at any level of the biliary tract and showing features of cholangiocellular differentiation [1,2]. Over the last few decades, epidemiological studies have revealed a steady increase of incidence and mortality from intrahepatic CCA (iCCA) globally, probably due to an increase of related risk factors, whereas the extrahepatic CCA (eCCA) frequency has slightly declined [3]. The American Cancer Society (www.cancer.org) reports that the 5-year overall survival rate is approximately 24% for iCCA and 13% for eCCA, respectively. In early stage CCA, surgical resection and liver transplantation are the only effective treatment options, yet most of CCA patients are diagnosed at advanced stages [4]. Due to the scarce understanding of CCA molecular pathogenesis, the range of therapeutic options available for progressed CCA is limited and unsatisfactory [1]. Indeed, the standard first line therapy for patients with unresectable CCA, consisting of the combined administration of gemcitabine and platin-based drugs, provides almost negligible benefits [5]. Thus, novel and more effective treatments against CCA are urgently needed.

Dysregulation of multiple kinases has been described as one of the major mechanisms by which cancer cells escape normal physiological constraints on proliferation and survival. Maternal embryonic leucine zipper kinase (MELK), a member of the snf1/AMP protein kinase family of serine/threonine kinases, activates multiple signaling cascades driving oncogenic growth [6,7]. MELK is a cell cycle-dependent kinase localized at chromosome 9p 13.2, whose expression is increased in many tumor tissues and in mitotically blocked cells, indicating that MELK expression is controlled during cell proliferation [8,9]. Various studies have shown that MELK interacts with a number of proteins, including the cell cycle protein CDC25B [10], mitogen-activated protein kinase (MAPK) [11], components of the Bcl-2 family [12], Smad proteins, which are intracellular mediators of the TGF-beta pathway [13], and TP53 [14]. Unlike other members of the AMPK family, MELK is not directly involved in the balance of cellular metabolism [7], but rather in cell cycle regulation, proliferation, apoptosis, and tumor formation [7,15]. MELK may also play a key role in tumor resistance to therapies and DNA repair, as MELK inhibition significantly increases the sensitivity to radiotherapy and chemotherapy in various models [16,17]. Thus, it is not surprising that high levels of MELK are detected in many solid tumors and in leukemia, where increased MELK expression correlates with poor prognosis and aggressiveness [18,19,20,21,22,23,24,25,26,27,28,29,30,31]. Genomic and pharmacological inhibition of MELK has been shown to hamper tumor growth, both in vitro and in various preclinical tumor models, further suggesting that this kinase could be a potential therapeutic target in cancer [22,28,29,32]. Although the molecular mechanisms whereby MELK induces aggressive tumor growth are not completely understood, it appears clear that MELK exerts its oncogenic role in association with the Forkhead Box Protein M1 (FOXM1) protooncogene [33], a master regulator of cell cycle progression that is remarkably activated in several human cancers, including HCC [21,34]. In the latter tumor type, MELK was found to be highly overexpressed and correlated to vascular invasion, early recurrence, and poor survival [21].

Moreover, after binding and phosphorylation of FOXM1, the MELK–FOXM1 activated complex binds to the promoter and induces the transcription of the enhancer of zeste homolog 2 (EZH2) gene [17,35]. Upregulation of EZH2, a lysine methyltransferase, leads to the transcriptional repression of genes promoting differentiation, while maintaining stem-like properties of cancer cells [17,24,36]. Some specific pathways containing MELK have been identified in glioblastoma multiforme (GBM), such as the c-JUN/MELK and MELK/PRC1 cascades, which support the survival of cancer stem cells (CSCs) [16]. Another study on GBM identified the protein Stathmin as a downstream target of the MELK pathway involved in the progression of this deadly disease [37]. In addition, a recent investigation revealed the eukaryotic translation initiation factor 4B (eIF4B) as a substrate of MELK. The latter is a novel signaling pathway that regulates protein synthesis during mitosis and, consequently, the survival of cancer cells [38]. Although mounting evidence indicates elevated levels of MELK in many tumor types, the molecular mechanisms responsible for the unrestrained activity of MELK remain poorly defined, and reports dealing with its regulation are scanty. A few previous studies suggested MELK as a downstream effector of the E2F1 transcription factor. Specifically, a genome-wide profiling of REF52 rat fibroblasts identified MELK as a member of a subset of E2F1 targets responsible for the anti-apoptotic properties of E2F1. In the same study, the authors also showed that MELK is upregulated in a subgroup of advanced, highly aggressive ovarian and breast tumors characterized by E2F1 overexpression [39]. In addition, a putative E2F1 binding site was identified in the MELK promoter in MC3T3-E1 mouse osteoblasts [40]. E2F1 is a member of the E2F family of transcription factors responsible for a plethora of biological functions, including cell cycle progression, apoptosis, DNA repair, and metabolism, and whose expression is frequently induced in cancer, leading to unrestrained growth and poor prognosis of the patients [41,42,43].

In light of these findings, suppression/inactivation of MELK may be an attractive therapeutic strategy in cancer. Currently, the only designated MELK inhibitor is OTSSP167, an orally administered drug, which is under evaluation in Phase 1 studies on various tumor types (identifiers: NCT01910545, NCT02926690, NCT02768519, and NCT02795520; www.clinicaltrials.gov) [44,45,46]. OTSSP167 binds to MELK, preventing both its phosphorylation and activation, thus inhibiting the phosphorylation of downstream MELK substrates [44,45,46]. It was reported that OTSSP167 reduced tumor burden in a murine model of multiple myeloma [45,46]. Specifically, OTSSP167 induced potent and rapid apoptosis of multiple myeloma cells, and impaired outgrowth of malignant plasma cells derived from myeloma stem cells in the peripheral blood [45,46]. Furthermore, OTSSP167 has been shown to suppress Sox2 expression in a dose-dependent manner in head and neck squamous cell carcinoma (HNSCC). Since Sox2 is involved in self renewal and tumorigenicity, MELK inhibition is believed to act as a key component of CSCs modulator through the regulation of Sox2 [47]. Also, OTSSP167 has been described as the most potent inhibitor of Human Cytomegalovirus (HCMV) replication in a study for the identification of already existing kinase inhibitors that can potentially be repurposed as novel antiviral drugs [48]. Nonetheless, kinobeads profiling and CRISPR/Cas9 mutagenesis studies recently demonstrated that OTSSP167 is a broad multi-kinase inhibitor. As a consequence of its off-target activity, the results of these studies undermined and did not support the clinical validation of OTSSP167 as a drug targeting MELK [49,50].

In the present study, we sought to determine the function of MELK in human intrahepatic cholangiocarcinoma (iCCA) by evaluating the levels of MELK and some of its targets in a collection of human iCCA and corresponding non-tumorous surrounding livers. Subsequently, we assessed the effects of MELK modulation on proliferation and apoptosis of iCCA cell lines. Finally, we investigated the role of E2F1 as an upstream inducer of MELK. Overall, our results indicate that MELK is an E2F1 effector gene potentially important as a prognostic marker and therapeutic target for human iCCA.

## 2. Materials and Methods

### 2.1. Human Tissue Samples

Five normal livers (obtained from residual liver tissues of liver donors), 52 iCCA, and corresponding surrounding non-tumor liver tissues were used for the study. Patients’ clinicopathological features are summarized in Table 1. Survival data were available for 39 patients. Liver tissues were collected at the Universities of Sassari (Sassari, Italy) and Greifswald (Greifswald, Germany). Institutional Review Board approval was obtained at local Ethical Committees of the Medical Universities of Sassari (approval code: 2377/CE; 31 May 2016) and Greifswald (approval code: BB 67/10; 3 June 2010), in compliance with the Helsinki Declaration. Written informed consent was obtained from all individuals.

### 2.2. Histology and Immunohistochemistry

Human liver specimens were fixed overnight in 4% paraformaldehyde and embedded in paraffin. Sections were done at 5 μm in thickness. Liver lesions were assessed by two board-certified pathologists and liver experts (S.R. and F.D.). For immunohistochemical staining, slides were deparaffinized in xylene, rehydrated through a graded alcohol series, and rinsed in PBS. Antigen retrieval was performed either in 10 mM sodium citrate buffer (pH 6.0) by placement in a microwave oven on high for 10 min, followed by a 20-min cool down at room temperature. After a blocking step with the 5% goat serum and Avidin-Biotin blocking kit (Vector Laboratories, Burlingame, CA, USA), the slides were incubated with the rabbit polyclonal anti-MELK (# HPA017214; Sigma-Aldrich, St. Louis, MO, USA; dilution 1:200), the rabbit monoclonal anti-EZH2 (# 5246; Cell Signaling Technology (Danvers, MA, USA); dilution 1:100), and the rabbit monoclonal anti-CK19 (# 12434; Cell Signaling Technology; 1:500) primary antibodies overnight at 4 °C. Slides were then subjected to 3% hydrogen peroxide for 10 min to quench endogenous peroxidase activity and, subsequently, the biotin conjugated secondary antibody was applied at a 1:500 dilution for 30 min at room temperature. The immunoreactivity was visualized with the Vectastain Elite ABC kit (Vector Laboratories, Burlingame, CA, USA) and Vector NovaRed (Vector Laboratories) as the chromogen. Slides were counterstained with hematoxylin.

### 2.3. Quantitative Reverse Transcription Real-Time Polymerase Chain Reaction (qRT-PCR)

Gene Expression Assays for human *MELK* (ID# Hs01106438_m1), *FOXM1* (Hs01073586_m1), *EZH2* (Hs00544830_m1), *E2F1* (ID # Hs00153451_m1), and *β-Actin* (ID # 4333762T) genes were purchased from Applied Biosystems (Foster City, CA, USA). Quantitative values for each gene were calculated by using the PE Biosystems Analysis software and expressed as number target (NT). NT = 2^−ΔCt^, wherein ΔCt value of each sample was calculated by subtracting the average Ct value of the target gene from the average Ct value of the *β-Actin* gene.

### 2.4. In Vitro Experiments

Mycoplasma-free, HUCCT1, and HUH28 human iCCA cell lines, after validation (Genetica DNA Laboratories, Burlington, NC, USA), were used for the in vitro studies. The two cell lines were kindly provided by Prof. Xin Chen (University of California San Francisco, San Francisco, CA, USA). Cell lines were maintained as monolayer cultures in Dulbecco’s modified Eagle medium supplemented with 10% fetal bovine serum (FBS; Gibco, Grand Island, NY, USA), 100 U/mL penicillin, and 100 g/mL streptomycin (Gibco, Grand Island, NY, USA). Cells were grown for 12 h, then serum-deprived for 24 h and treated with siRNA against *MELK* (# S386; ThermoFisher Scientific) or *E2F1* (# S4405; ThermoFisher Scientific), following the manufacturer’s recommendations. In addition, HUCCT1 and HUH28 cells were seeded in 24-well plates and treated with the non-selective MELK inhibitor OTSSP167 (Selleck Chemicals, Houston, TX, USA) at 40 and 100 nM concentration for 24–48 h. For the induction of DNA damage, cells were treated with 2 μmol/L doxorubicin (Sigma-Aldrich, St. Louis, MO, USA) for 2 h, washed twice with phosphate-buffered saline (PBS), and returned to normal growth medium for the indicated time periods. Cell proliferation was assessed in the cell lines at 24- and 48-h time points using the BrdU Cell Proliferation Assay Kit (Cell Signaling Technology, Danvers, MA, USA). As concerns apoptosis, it was determined in the two iCCA cell lines using the Cell Death Detection Elisa plus Kit (Roche Molecular Biochemicals, Indianapolis, IN, USA), following the manufacturer’ instructions. Cells were initially subjected to 24 h serum starvation. Subsequently, apoptotic cell death was assessed at 24- and 48-h time points. Finally, to assess DNA damage of cultured HCC cells, the DNA Damage Quantification Kit (Biovision, Mountain View, CA, USA) was applied following the manufacturer’s instructions, and results were recorded at 24- and 48-h time points. All cell line experiments were repeated at least three times in triplicate.

### 2.5. Statistical and Bioinformatic Analysis

GraphPad Prism version 6.0 (GraphPad Software Inc., La Jolla, CA, USA) was used to evaluate statistical significance by Tukey–Kramer, Student’s *t,* and Mann–Whitney tests and linear regression analyses. Values of *p* < 0.05 were considered significant. Data are expressed as mean ± standard deviation for each group. Two-tailed unpaired *t*-test was used to compare the differences between two groups.

*E2F1* putative transcription factor binding motifs on *MELK* gene promoters were predicted using the Eukaryotic Promoter Database New (EPDnew; https://epd.epfl.ch) that combines EPD promoters with promoter-specific high-throughput data [51].

## 3. Results

### 3.1. Overexpression of MELK in Human Intrahepatic Cholangiocarcinoma

First, we investigated the mRNA levels of *MELK* gene by real-time RT-PCR in an iCCA collection (*n* = 52; Table 1). Strikingly, significantly higher *MELK* mRNA levels when compared with corresponding non-tumorous counterparts (ST) and normal livers (NL) were detected (Figure 1A). A similar trend of expression was detected when assessing the levels of two canonical MELK targets, namely *Forkhead Box Protein M1* (*FOXM1*; Figure 1B) and *Enhancer of Zeste Homolog 2* (*EZH2*; Figure 1C) genes. Consequently, a significant correlation between the mRNA levels of *MELK* and those of *FOXM1* and *EZH2* was observed (Figure 1D–F). Subsequently, to further substantiate our data, we assessed the protein levels of MELK and EZH2 by way of immunohistochemistry in the same iCCA collection. In normal healthy liver and non-tumorous surrounding liver tissues, low cytoplasmic and weak or absent nuclear immunolabeling for MELK was observed (Figure 2). Faint nuclear immunolocalization for the EZH2 protein was detected in the same tissues. In striking contrast and in agreement with real-time RT-PCR data, we found a strong cytoplasmic and/or nuclear immunoreactivity for MELK and EZH2 proteins in iCCA (Figure 3).

Overall, these concordant findings indicate the simultaneous upregulation of MELK and its targets (FOXM1 and EZH2) in human iCCA specimens.

### 3.2. MELK is a Negative Prognostic Indicator in Intrahepatic Cholangiocarcinoma

To determine a possible prognostic role of MELK in this tumor type, levels of *MELK* mRNA in iCCA were related to the length of survival of the patients. Survival data were available for 39 of the 52 patients examined. Noticeably, we found that higher *MELK* gene expression correlates with lower iCCA survival rate, as assessed both by Kaplan–Meier and linear regression analysis (*p* < 0.0001; Figure 4A,D). This association remained strongly significant after multivariate Cox regression analysis (*p* < 0.0001; Appendix A), thus indicating that *MELK* mRNA levels represent an independent prognostic factor for iCCA. Equivalent results were observed when assessing the correlation between *FOXM1* (Figure 4B,E) or *EZH2* (Figure 4C,F) expression and the length of patients’ survival using the same statistical tools. No association between the levels of *MELK* mRNA and clinicopathologic features of iCCA patients, including age, gender, etiology, presence of cirrhosis, tumor size, and tumor differentiation, was detected (Appendix A).

Taken together, the present findings indicate that induction of the *MELK* gene is almost universal in human iCCA, where *MELK* overexpression predicts poor outcome for iCCA patients.

### 3.3. Suppression of MELK is Highly Detrimental for the Growth of Human iCCA Cell Lines

Subsequently, we evaluated the effects of MELK suppression on the growth of iCCA. For this purpose, we assessed the growth inhibitory activity of the MELK non-selective inhibitor OTSSP167 on human iCCA cell lines. The HUCCT1 and HuH28 iCCA cell lines were selected for the experiments (Figure 5). At the cellular level, OTSSP167 resulted in decreased proliferation and rise of apoptosis in the two cell lines in a dose-dependent manner. Due to the established role of MELK in promoting DNA repair and genome integrity [16,17], we hypothesized that cancer cells depleted of MELK might be more vulnerable to DNA insults than the corresponding cells retaining MELK activity. To test this hypothesis, the two iCCA cell lines were subjected to OTSSP167 treatment in association with doxorubicin, a DNA damaging agent. Strikingly, an impressive growth restraint, elevated apoptosis, and massive DNA damage were detected in HUCCT1 and HuH28 cells when administration of OTSSP167 was coupled to treatment with doxorubicin. At the molecular level, treatment with OTSSP167 did not affect *MELK* mRNA levels in HUCCT1 and HuH28 cell lines, in accordance with the notion that OPSSP167 impairs MELK activity but not its expression (Figure 6) [44]. In contrast, downregulation of *FOXM1* and *EZH2* gene expression was detected following OTSSP167 administration in both cell lines. Of note, doxorubicin treatment induced the upregulation of *MELK*, *FOXM1*, and *EZH2* genes, in accordance with the role of MELK in the DNA repair response (Figure 6) [16,17].

Next, to exclude possible off-target effects of OTSSP167 [44,45], the HUCCT1 and HUH28 cell lines were subjected to *MELK* knockdown using specific small interfering RNA (siRNA) against *MELK* (Figure 7). Silencing of *MELK* resulted in slight decrease of proliferation and induction apoptosis in the two iCCA cell lines. Similar to that described for the OTSSP167 treatment (although less pronounced), a strong growth reduction and generation of DNA damage was induced when combining doxorubicin administration to *MELK* knockdown. As expected, at the molecular level (Figure 8), treatment with *MELK* siRNA resulted in the downregulation of *MELK* and its downstream effectors, *FOXM1* and *EZH2*.

Altogether, the present data indicate that suppression of MELK significantly constrains the growth of iCCA cell lines in vitro, especially in association with DNA damaging agents.

### 3.4. MELK Is a Transcriptional Target of the E2F1 Protooncogene in Intrahepatic Cholangiocarcinoma

Finally, we sought to determine whether MELK is a downstream effector of the E2F1 transcription factor in iCCA. Several approaches were employed for this purpose. First, we performed an in silico prediction of the binding sites for E2F1 on *MELK* gene promoter using the EPDnew software [51]. Noticeably, we identified two putative binding sites for E2F1 on the *MELK* gene promoter region (*p* < 0.01; Figure 9A), situated −119 and +81 from the ATG starting codon. Next, *E2F1* was knocked down in the HUCCT1 and HUH28 cell lines by specific siRNA. As expected, siRNA-mediated silencing of *E2F1* resulted in both cell lines in a marked downregulation of *MELK*, further substantiating the hypothesis of MELK being a transcriptional target of E2F1 (Figure 9B,C). Subsequently, we examined whether a correlation between *MELK* and *E2F1* mRNA levels exist in our iCCA collection. The trend of *E2F1* gene expression in the iCCA collection closely recapitulated that of *MELK*, with highest levels being detected in iCCA and lower levels in normal livers and non-tumorous surrounding livers. No significant differences were observed between the last two entities (Figure 9D). As a consequence, a strong direct correlation between the levels of the two transcription factors was found (Figure 9E).

Overall, the present results identify E2F1 as an upstream inducer of MELK in iCCA.

## 4. Discussion

Intrahepatic cholangiocarcinoma (iCCA) is a highly lethal tumor, characterized by rising incidence, clinical aggressiveness, high recurrence rate, and limited therapeutic options [1,2,3,4,5]. The prognosis of iCCA patients is most often poor due the late diagnosis of the disease and the lack of effective treatments when the tumor is unresectable [1,2,3,4,5]. To overcome this gloomy scenario, novel molecular targets and more effective therapies are urgently needed.

In the present investigation, we focused on the role in cholangiocarcinogenesis of MELK, a gene frequently activated in a variety of tumor types [6]. We show here that MELK is overexpressed and possesses oncogenic properties in human iCCA. Indeed, we found a significant upregulation of MELK and its targets, FOXM1 and EZH2, in iCCA when compared with normal livers and non-tumorous surrounding livers. These findings suggest that induction of MELK might be involved in liver malignant transformation. Furthermore, a marked increase of MELK expression was detected in tumors associated with shorter survival length of the patients, thus underlining a role of MELK also in iCCA biological aggressiveness and the patient’s prognosis. In agreement with our data, high levels of MELK have been previously shown to directly correlate with an unfavorable outcome in various cancer entities [18,19,20,21,22,23,24,25,26,27,28,29,30,31]. Therefore, overall the present findings strongly support an important pathogenetic and prognostic role of MELK in human iCCA.

In in vitro experiments, we found that MELK suppression by way of the non-selective inhibitor OTSSP167 or specific siRNA significantly reduced the proliferation and augmented apoptosis in iCCA cells, suggesting that MELK contributes to cholangiocarcinogenesis via multiple mechanisms. Of note, the deleterious effects on cell growth were markedly amplified by the treatment with the DNA-damaging agent doxorubicin. These data suggest a major function of MELK in protecting the iCCA cells from DNA damage, in agreement with previous data obtained in glioma cells [16,17,24]. The molecular mechanisms whereby MELK promotes the DNA repair response remain poorly understood. It has been previously shown that MELK can bind to and phosphorylate the p53 tumor suppressor [14], suggesting that MELK properties on preserving the DNA integrity might be p53-dependent. However, this at least partly does not hold true for iCCA, since both cell lines used in this study (HUCCT1 and HUH28) harbor mutations in the p53 gene. Further studies are necessary to determine the precise mechanisms through which MELK contributes to the DNA damage response.

Although both approaches aimed at inhibiting MELK activity (namely OTSSP167 administration and siRNA-mediated *MELK* gene silencing) were able to reduce the growth of HUCCT1 and HUH28 cell lines, a stronger anti-neoplastic activity was observed when administering the non-selecting inhibitor to the cells. Since it has been proven that OTSSP167 does not selectively inhibit MELK [49,50], we cannot exclude that part of the anti-growth function (s) of OTSSP167 is due to off-target effects of the drug. In support of the latter hypothesis, a recent investigation showed that OTSSP167 hampers the proliferation and colony formation ability of lung cancer cell lines, as well as carcinogenesis in lung cancer xenografts via the suppression of the PAK1 protooncogene [52]. Furthermore, it has been found that *MELK* suppression by siRNA did not recapitulate the mitotic effects of OTSSP167 treatment in MCF7 and Hela cancer cell lines. In these cells, the growth inhibitory activity of the drug in fact mainly depended on the inhibition of other kinases, such as BUB1 and Haspin [53]. Thus, to address the functions of MELK in cancer more precisely, either MELK genetically engineered mouse models and/or selective inhibitors of this protein should be used. In this regard, the small-molecule inhibitor MELK-T1 has been recently developed [54] and might be highly helpful for this purpose. Nonetheless, the data obtained in the present study and in many other tumor types support further investigation on the effectiveness of OTSSP167 as an anticancer drug and underline the importance of ongoing clinical trials testing this drug in various tumor entities (NCT01910545, NCT02926690, NCT02768519, and NCT02795520; www.clinicaltrials.gov).

At the molecular level, we identified the E2F1 transcription factor as the upstream inducer of MELK in iCCA. E2F1 is a major player in various physiological and pathological events, due to its ability to regulate a wide number of cellular events, including cell cycle, apoptosis, DNA repair, and metabolism [41,42,43,55,56,57,58,59]. Of note, the E2F1 transcriptional program is aberrantly activated in the vast majority of human iCCA due to the inactivation of the retinoblastoma protein (pRb) and is sensitive to the treatment with CDK4/6 inhibitors such as Palbociclib [60]. Thus, it is possible that MELK represents a surrogate marker of E2F1 activation in this tumor type and might be helpful to select people to be treated with CDK4/6 inhibitors.

## 5. Conclusions

In summary, we revealed that MELK is an E2F1-responsive target presumably playing a crucial role in human iCCA development and progression. Also, we envisaged the possibility that MELK is a prognostic marker for this tumor type. Furthermore, we found that the anti-growth effects driven by MELK inhibition are strongly amplified by the concomitant administration of a DNA damaging agent. Thus, the administration of MELK inhibitors, together with DNA damaging agents, might represent a novel and promising therapeutic strategy against human iCCA.

## Figures and Tables

**Figure 1 medicina-56-00001-f001:**
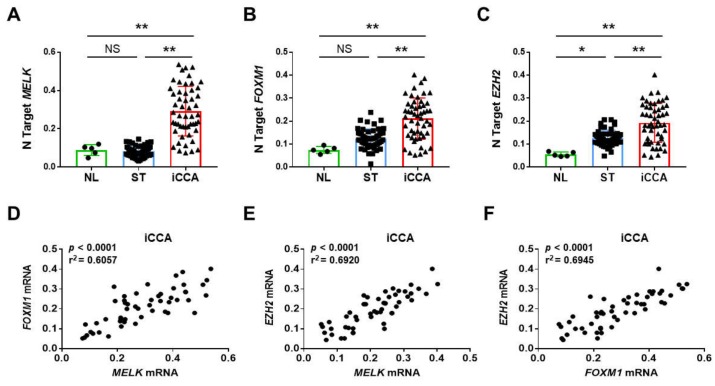
MELK and its downstream effectors are overexpressed in human intrahepatic cholangiocarcinoma (iCCA) specimens. (**A**–**C**) Quantitative real-time RT-PCR analysis of *MELK*, *FOXM1*, and *EZH2* mRNA levels in normal livers (*n* = 5), iCCA (*n* = 52), and corresponding non-tumorous surrounding liver tissues (ST; *n* = 52). Quantitative values were calculated by using the PE Biosystems Analysis software and expressed as number target (N Target). N Target = 2^−ΔCt^, wherein the ΔCt value of each sample was calculated by subtracting the average Ct value of the gene of interest from the average Ct value of the *β-Actin* gene. *p*-value was calculated using Mann–Whitney *U* test. * *p* < 0.01 when compared to Normal Liver (NL); ** *p* < 0.001 when compared to Normal liver (NL) and Surrounding Tissue (ST). Abbreviation: NS, not significant. (**D**–**F**) Expression levels of *MELK*, *FOXM1*, and *EZH2* correlate with each other in human iCCA samples.

**Figure 2 medicina-56-00001-f002:**
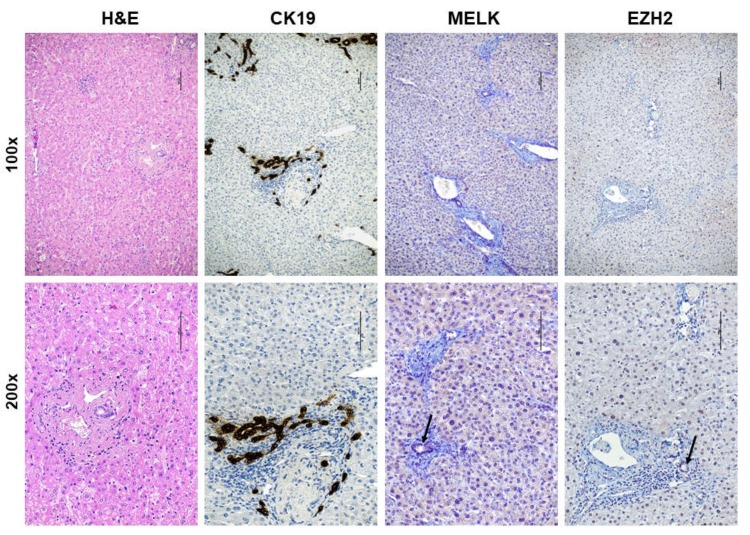
Representative immunohistochemical patterns of MELK and EZH2 proteins in human non-tumorous surrounding tumor tissues. A non-neoplastic portion of liver tissue in the proximity of one intrahepatic cholangiocarcinoma is displayed at two magnifications (100×, upper panels; 200×, lower panels) and shows weak cytoplasmic and nuclear staining of hepatocytes for MELK and EZH2 proteins, respectively. Low to moderate immunoreactivity for MELK and EZH2 in biliary epithelial cell is indicated by arrows. An equivalent staining pattern for MELK and EZH2 characterized normal healthy liver tissues (not shown). The CK19 staining was used as a marker of biliary differentiation of the tumors. Scale bar: 100 μm. Abbreviation: H&E, hematoxylin and eosin staining.

**Figure 3 medicina-56-00001-f003:**
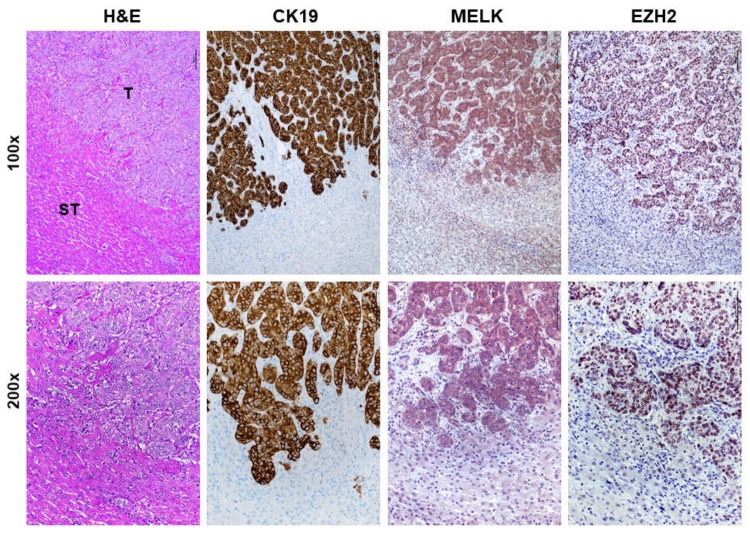
Representative immunohistochemical patterns of MELK and EZH2 proteins in human intrahepatic cholangiocarcinoma specimens (iCCA). A cholangiocellular tumor (T) is shown at two magnifications (100×, upper panels; 200×, lower panels) and exhibits concomitantly strong cytoplasmic and/or nuclear staining for MELK and EZH2 proteins, whereas weak/absent immunoreactivity is displayed by the non-tumorous surrounding counterpart (ST). The CK19 staining was used as a marker of biliary differentiation of the tumors. Scale bar: 100 μm. Abbreviation: H&E, hematoxylin and eosin staining.

**Figure 4 medicina-56-00001-f004:**
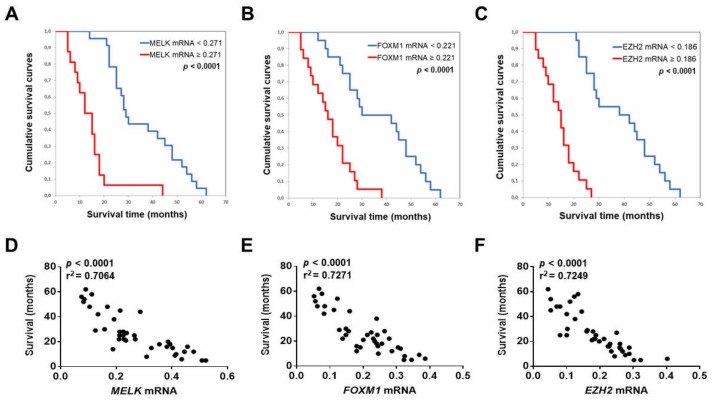
MELK, FOXM1, and EZH2 levels correlate with an adverse outcome in human intrahepatic cholangiocarcinoma (iCCA). (**A**–**C**) Kaplan–Meier survival curves of human iCCA with high and low *MELK*, *FOXM1*, and *EZH2* mRNA levels, showing the unfavorable outcome of patients with elevated expression of the three genes. (**D**–**F**) Linear regression analysis showing a statistically significant, inverse correlation between the levels of *MELK*, *FOXM1*, and *EZH2*, and the length (months) of patients’ survival.

**Figure 5 medicina-56-00001-f005:**
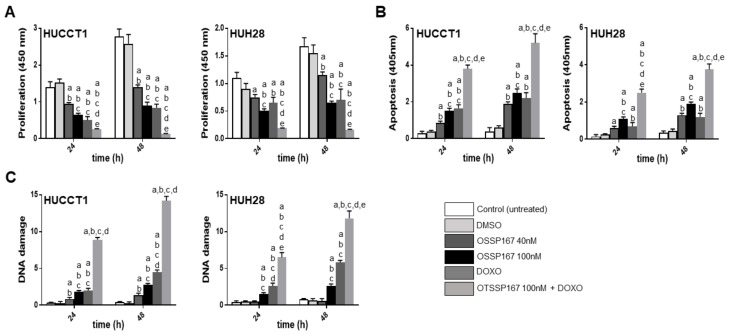
Effect of inhibition of MELK by the non-selective inhibitor OTSSP167, either alone or in association with doxorubicin administration (DOXO) on proliferation (A; OD 450 nm), apoptosis (B; OD 405 nm), and DNA damage (C; absolute number of apurinic sites per 10^5^ base pairs) of HUCCT1 and HUH28 intrahepatic cholangiocarcinoma (iCCA) cell lines. Data are means ± standard deviation (SD) of three experiments conducted in triplicate. Tukey–Kramer test: At least *p* < 0.01; a, vs. control (untreated cells); b, vs. DMSO (solvent); c, vs. OTSSP167 40 nM; d, vs. OTSSP167 100 nM; e, vs. DOXO.

**Figure 6 medicina-56-00001-f006:**
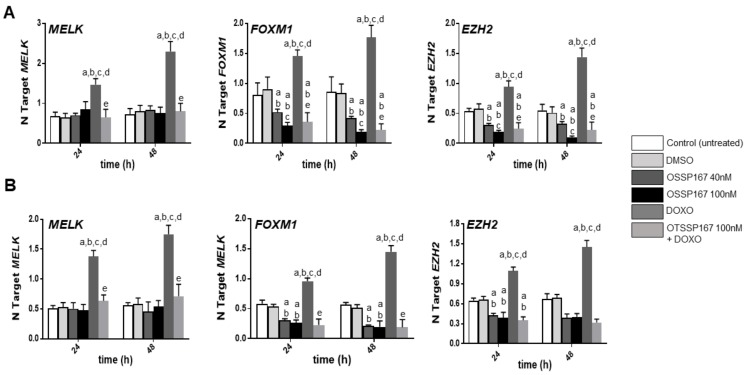
Effect of inhibition of MELK activity by the non-selective inhibitor OTSSP167, either alone or in association with doxorubicin administration (DOXO) on the levels of *MELK*, *FOXM1*, and *EZH2* mRNA in HUCCT1 (**A**) and HUH28 (**B**) human intrahepatic cholangiocarcinoma (iCCA) cell lines. Data are means ± SD of three experiments conducted in triplicate. Tukey–Kramer test: At least *p* < 0.01; a, vs. control (untreated cells); b, vs. DMSO (solvent); c, vs. OTSSP167 40 nM; d, vs. OTSSP167 100 nM; e, vs. DOXO.

**Figure 7 medicina-56-00001-f007:**
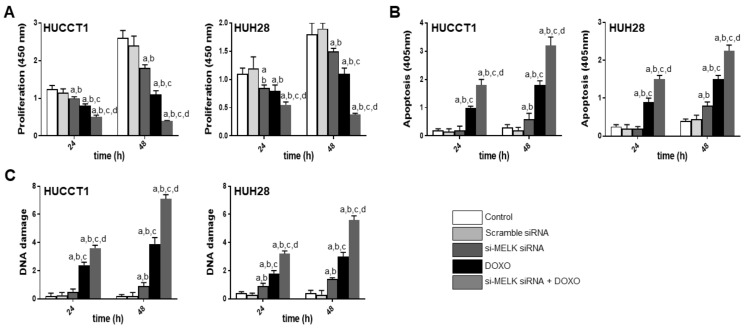
Effect of inhibition of MELK expression by the specific siRNA against MELK, either alone or association with doxorubicin administration (DOXO) on proliferation (A; OD 450 nm), apoptosis (B; OD 405 nm), and DNA damage (C; absolute number of apurinic sites per 10^5^ base pairs) of HUCCT1 and HUH28 intrahepatic cholangiocarcinoma (iCCA) cell lines. Data are means ± SD of three experiments conducted in triplicate. Tukey–Kramer test: At least *p* < 0.01; a, vs. control (untreated cells); b, vs. scramble siRNA; c, vs. *MELK* siRNA; d, vs. DOXO.

**Figure 8 medicina-56-00001-f008:**
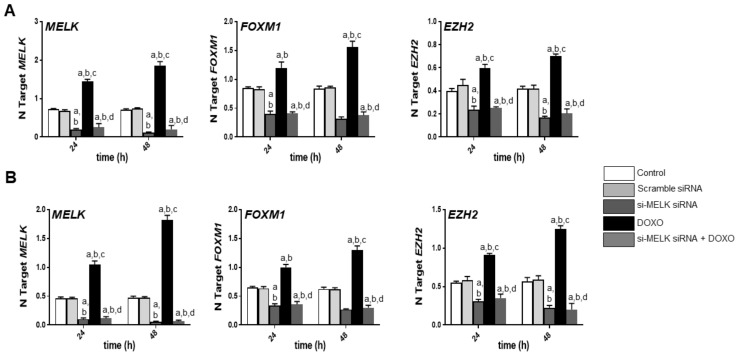
Effect of inhibition of *MELK* expression by the specific siRNA against *MELK*, either alone or association with doxorubicin administration (DOXO) on the levels of *MELK*, *FOXM1*, and *EZH2* mRNA in HUCCT1 (**A**) and HUH28 (**B**) human intrahepatic cholangiocarcinoma (iCCA) cell lines. Tukey–Kramer test: At least *p* < 0.001; a, vs. control (untreated cells); b, vs. scramble siRNA; c, vs. *MELK* siRNA; d, vs. DOXO.

**Figure 9 medicina-56-00001-f009:**
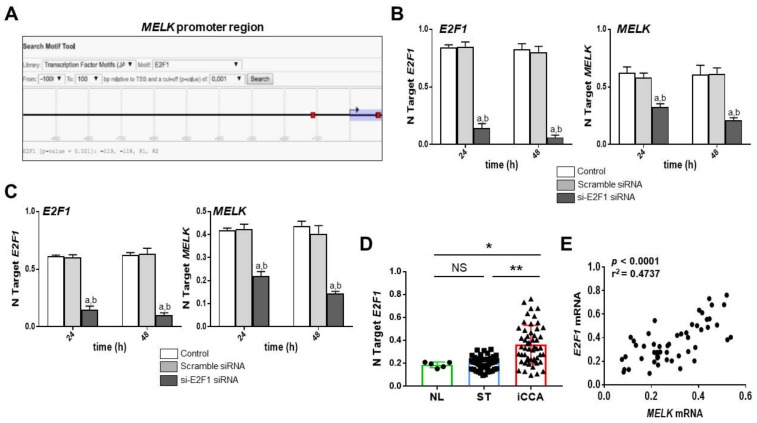
E2F1 is an upstream inducer of MELK in human intrahepatic cholangiocarcinoma (iCCA) cell lines. (**A**) In silico prediction of E2F1 binding sites on the *MELK* gene promoter using a cut-off of *p*-value of 0.001. Binding sites are represented in red squares. The EPDnew software was used for this scope. (**B**) Inhibition of *E2F1* expression (left panel) by the specific siRNA against *E2F1* induces the downregulation of *MELK* mRNA levels (right panel) in the HUCCT1 cell line. (**C**) Inhibition of *E2F1* expression (left panel) by the specific siRNA against *E2F1* triggers the downregulation of *MELK* mRNA levels (right panel) in the HUH28 cell line. Data are means ± SD of three experiments conducted in triplicate. Tukey–Kramer test: At least *p* < 0.001; a, vs. control (untreated cells); b, vs. scramble siRNA. (**D**) Quantitative real-time RT-PCR analysis of *E2F1* mRNA levels in normal livers (*n* = 5), iCCA (*n* = 52), and corresponding non-tumorous surrounding liver tissues (ST; *n* = 52). Quantitative values were calculated by using the PE Biosystems Analysis software and expressed as number target (N Target). N Target = 2^−ΔCt^, wherein the ΔCt value of each sample was calculated by subtracting the average Ct value of *E2F1* gene from the average Ct value of the *β-Actin* gene. *p*-value was calculated using Mann–Whitney *U* test. ** *p* < 0.001 when compared to ST; * *p* < 0.01 when compared with NL. Abbreviation: NS, not significant. (**E**) Expression levels of *E2F1* and *MELK* correlate with each other in human iCCA samples.

**Table 1 medicina-56-00001-t001:** Clinicopathological features of intrahepatic cholangiocarcinoma (iCCA) patients.

Variables	
No. of patients; male; female	52; 31; 21
Age (years); <60; >60	14; 38
Etiology; HBV; HCV; hepatolithiasis; PSC; NA	12; 7; 12; 3; 18
Liver cirrhosis; yes; no	20; 32
Tumor differentiation; well; moderately; poorly	24; 20; 8
Tumor size (cm); <5; >5	36; 16
Tumor number; single; multiple	38; 14
Lymph node metastases; yes; no; NA	19; 21; 12

Abbreviations: NA, not available; PSC, primary sclerosing cholangitis.

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
