# Peer review of "Inhibition of MELK Protooncogene as an Innovative Treatment for Intrahepatic Cholangiocarcinoma"

_medicina, 2019, doi:10.3390/medicina56010001_

Round 1

Reviewer 1 Report

In this manuscript, Cigliano and colleagues studied the expression and relevance of MELK and of two its downstream effectors (FOXM1 and EZH2) in intrahepatic cholangiocarcinoma (iCCA); Authors analyzed this both in vitro and on a quite large cohort of patients. Authors evaluated the expression of the three target proteins using real time PCR and immunohistochemistry. The most important results reached by the Authors are that a. MELK expression, as well as of FOXM1 and EZH2, are significantly higher in iCCA human samples respect to both normal controls (NL), and perilesional tissue (ST); b. the expression of these three markers is related to a worst outcome of the patients; c. the treatment in vivo with OTSSP167, an inhibitor of MELK, reduces proliferation and stimulates apoptosis in two different intrahepatic cell lines (HuCCT1 and Huh28); d. MELK is a downstream effector of E2F1 transcription factor.

The paper is correctly written and data are novel and of interest, despite this, some flows significantly dampen the enthusiasm for this paper.

E2F1 come out of the blue; please discuss thoroughly the importance and significance of this transcription factor in the background and discussion sections. IHC expression of the different proteins in NL and ST, and not only in iCCA samples (Fig. 2) could be showed, at least as supplemental figure. Basal expression of MELK, FOXM1, and EZH2 on iCCA cells was showed only in Fig. 6, these data are fundamental for the experiment showed in figs 4 and 5, and must be anticipated. References have to be edited

Author Response

We sincerely thank the Reviewer for the insightful observations.

- In agreement with the Reviewer, we have modified the Introduction and Discussion sections of the manuscript in order to highlight the importance of the E2F1 transcription factor in cancer and added some appropriate references on this topic.

- The immunohistochemistry in former Figure 2 already shows the clear difference in staining intensity for MELK and EZH2 proteins between the tumor and the adjacent non-tumorous compartment, as the pictures were taken at the tumor edge. Nevertheless, in accordance with the Reviewer, we have added a new figure (new Figure 2) showing in more detail the staining patterns for MELK and EZH2 proteins in surrounding non-tumorous liver. The staining patter for the two proteins in the non-tumorous surrounding tissue is identical to that identified in healthy normal liver; consequently, the latter is not shown.

- Expression levels of MELK, FOXM1, and EZH2 genes in HUCCT1 and HuH28 cell lines following the treatment with OTSSP167, either alone or in association with the DNA damaging agent Doxorubicin (Figure 4), were mistakenly omitted in the original version of the manuscript. They have been included in the revised version of the manuscript as (new) Figure 5 and the results described in the text. Levels of the same genes for the experiments depicted in former Figure 5 (now Figure 6 in the revised manuscript) are shown in Figure 7.

- The references have been edited as requested.

Reviewer 2 Report

Intrahepatic cholangiocarcinoma (iCCA) is a pernicious tumor characterized by a dismal outcome and scarce therapeutic options. The present study was aimed at elucidating the role of the maternal embryonic leucine zipper kinase (MELK) protooncogene in iCCA. The effects on iCCA growth of boththe multi-kinase inhibitor OTSSP167 and specific small-interfering RNA (siRNA) against MELK were investigated in iCCA cell line. Results indicate that MELK is ubiquitously overexpressed in iCCA, where it may represent a prognostic indicator and a therapeutic target.

This is an important and well written paper. Unlike extrahepatic CC which are more accessible and easier to resect, iCCAs involve bulk liver making their surgical management more challenging. Results are clearly presented. Is it possible to plot correlation between MELK mrNA vs. length of survival for the 39 patients in addition to the KM plot in Figure 3?  

Minor

line 49   Deregulation of multiple kinases – change to dysregulation

Author Response

We thank the Reviewer for the appreciation of our manuscript.

Specific issues:

1.) “Is it possible to plot correlation between MELK mRNA vs. length of survival for the 39 patients in addition to the KM plot in Figure 3?

Response: In agreement with the Reviewer, we have investigated the correlation between survival length and mRNA levels of MELK, FOXM1 and EZH2 genes by linear regression analysis (please refer to revised Figure 3, panels D-F). The results obtained by the analysis clearly show the existence of an inverse, significant correlation between the levels of the three genes and the length of patients’ survival.

2.) “Line 49 Deregulation of multiple kinases – change to dysregulation

Response: The text has been modified according to the suggestion of the Reviewer.

Round 2

Reviewer 1 Report

Authors satisfactorily answered the question raised by the reviewer; the paper is now suitable for publication.